# COVID-19 Pandemic-Related Impacts on Newborn Screening Public Health Surveillance

**DOI:** 10.3390/ijns8020028

**Published:** 2022-04-15

**Authors:** Sikha Singh, Michele Caggana, Carol Johnson, Rachel Lee, Guisou Zarbalian, Amy Gaviglio, Alisha Keehn, Mia Morrison, Scott J. Becker, Jelili Ojodu

**Affiliations:** 1Association of Public Health Laboratories, Silver Spring, MD 20910, USA; guisou.zarbalian@aphl.org (G.Z.); amy.gaviglio@aphl.org (A.G.); scott.becker@aphl.org (S.J.B.); jelili.ojodu@aphl.org (J.O.); 2Wadsworth Center, New York State Department of Health, Newborn Screening Program, David Axelrod Institute, Albany, NY 12201, USA; michele.caggana@health.ny.gov; 3Stead Family Children’s Hospital, University of Iowa, Iowa City, IA 52242, USA; carol.johnson@uiowa.edu; 4Laboratory Services Section, Texas Department of State Health Services, Austin, TX 78756, USA; rachel.lee@dshs.texas.gov; 5U.S. Department of Health and Human Services, Health Resources and Services Administration, Maternal and Child Health Bureau, Rockville, MD 20857, USA; akeehn@hrsa.gov (A.K.); mmorrison@hrsa.gov (M.M.)

**Keywords:** newborn screening, COVID-19, public health, NewSTEPs

## Abstract

Newborn screening (NBS) is an essential public health service that performs screening to identify those newborns at increased risk for a panel of disorders, most of which are genetic. The goal of screening is to link those newborns at the highest risk to timely intervention and potentially life-saving treatment. The global COVID-19 pandemic led to disruptions within the United States public health system, revealing implications for the continuity of newborn screening laboratories and follow-up operations. The impacts of COVID-19 across different states at various time points meant that NBS programs impacted by the pandemic later could benefit from the immediate experiences of the earlier impacted programs. This article will review the collection, analysis, and dissemination of information during the COVID-19 pandemic facilitated by a national, centralized technical assistance and resource center for NBS programs.

## 1. Introduction

The Association of Public Health Laboratories (APHL) works to strengthen public health laboratory systems, and during public health emergencies, it supports laboratory response [1]. During the novel coronavirus (COVID-19) pandemic, the newborn screening and genetics program of APHL served as the national coordinating center, as well as an information dissemination resource, to newborn screening laboratories and follow-up programs across the United States. Newborn screening (NBS) enables early detection and treatment of rare disorders by testing approximately 3.6 million babies [2] in the US each year and identifying over 12,000 positive cases [3]. NBS is an essential life-saving service that must continue despite public health emergencies even when other public services are slowed or halted.

In January 2020, the first cases of COVID-19 were detected in the United States, and on 22 January 2020, APHL initiated its Incident Command System (ICS) to manage planning and response to the outbreak and to ensure continuity of public health program activities. By February 2020, the United States was detecting community transmission of the virus, by mid-March, all 50 states and four US territories had reported cases of COVID-19 [4], and on 11 March 2020, the World Health Organization (WHO) declared a global pandemic [5].

Public Health Laboratories continues to play a key role in surveillance and testing for the COVID-19 pandemic, and this impacts routine NBS operations. In early 2020, competing priorities, coupled with the necessity to implement contingencies in each state, revealed unique, ongoing challenges for the NBS program. These challenges were met with solutions to the rapid shifts during the pandemic in order to maintain NBS continuity.

The Newborn Screening Technical assistance and Evaluation Program (NewSTEPs) of APHL, as well as federal partners, state stakeholders, and national organizations, coordinated efforts to ensure that communication, information sharing, just-in-time resources, and funding remained available to state NBS programs, with the goal of supporting their ongoing operations and evolving needs during the public health emergency during a time when staffing, resources, and physical space were compromised. Table 1 summarizes internal and external key events and activities impacting newborn screening public health surveillance in the US during the global pandemic. The goal of this analysis is to review the impacts of the COVID-19 pandemic on newborn screening. Additionally, we examine the utility of a national, centralized resource center for data collection and information dissemination.

## 2. Methods

### 2.1. Key Resources in COVID-19 Response

#### 2.1.1. National Technical Assistance Center

NewSTEPs, funded by the Maternal and Child Health Bureau (MCHB) of the Health Resources and Services Administration (HRSA), serves as the national, federally funded technical assistance resource center for newborn screening laboratory and follow-up programs, centralizing information collection, collation, and dissemination efforts [6]. In the initial months of the pandemic, NewSTEPs quickly developed, deployed, and made daily updates to a COVID-19 NBS Practices and Resources website, geared toward an audience of state NBS laboratory and follow-up program staff [7]. The website displayed information from state newborn screening programs collected through communications NewSTEPs staff and was stratified by the following topics: Continuity of Operations in a Pandemic, Supply Shortages, Disorder Specific Response, Courier Challenges, Education and Outreach, Second/Repeat Screens, Telehealth, Continuity of Operations Plans, and Biosafety of Specimens. Each topic area featured the challenges being experienced relative to each component of the NBS system areas, as well as resources and strategies sourced directly from state NBS programs and national newborn screening stakeholders such as the Genetic Alliance and the Association of Maternal and Child Health Programs (AMCHP). An analysis of provider impacts has been recently published by colleagues in Philadelphia, USA [8]. Initially, only a handful of states experienced the brunt of the pandemic, and the NBS programs within these states were able to share information about how they adapted through the centralized NewSTEPs website. Due to this, other peer NBS programs were able to anticipate and react to the pandemic impacts felt by these early hit states.

#### 2.1.2. National Webinars

Beginning in May 2020, NewSTEPs hosted national webinars through which state NBS programs and federal partners presented just-in-time content that was recorded and made freely available throughout the course of the COVID-19 pandemic [9]. Topics were relevant and focused on state-led solutions to emerging issues (Table 2).

#### 2.1.3. Public Health Emergency Funding

In September 2020, the HRSA MCHB provided supplemental federal funding to APHL and, in turn, APHL issued a Request for Proposal (RFP) to all state NBS programs focused on addressing the impact of the public health emergency on NBS [10]. APHL funded all seven applicants to strengthen their NBS operations during the COVID-19 public health emergency. The limited number of applicants was a reflection of the competing priorities of state NBS programs during this time, as well as of the funding level.

#### 2.1.4. Telehealth

The Coronavirus Aid, Relief, and Economic Security Act or CARES ACT allotted funding from HRSA to the Association of Maternal and Child Health Programs (AMCHP) in the summer of 2020 and resulted in a collaboration between AMCHP and APHL to support state NBS programs with telehealth focused activities and resource development [11].

### 2.2. Data in COVID-19 Response

#### 2.2.1. Survey Instrument

Discussions at various levels of APHL NBS committees and subcommittees revealed anecdotally that the pandemic was affecting NBS laboratories and follow-up programs differently, even within the same state. In November 2020, APHL fielded a survey using the SurveyMonkey platform to all state and territorial NBS programs, allowing multiple survey responses per program. Following the three-week survey period, 45 survey responses were received, representing 34 NBS programs. The survey sought to understand the spectrum of effects across NBS programs on staff, Information Technology resources, specimen transport, reagent and/or supply shortages, and support from vendors.

#### 2.2.2. Quality Metrics and Data Collection

The existing centralized NewSTEPs Data Repository collects harmonized metrics from all US NBS programs that have a signed data use agreement with APHL with the intent of comparing quality practices across and within states. The data are subject to privacy and security requirements, with state-specific data shared only in aggregate and subject to consent to release. The quality metrics collected by NewSTEPs cover pre-analytic, analytic, and post-analytic NBS practices, including factors such as specimen quality, timeliness of collection and reporting, missed cases, and presumptive and confirmed positive cases [12]. NewSTEPs became the one-stop resource center for COVID-related information for the NBS laboratory and follow-up programs, and the data collection efforts will support retrospective data analysis.

APHL established its Incident Command System (ICS) and activated its Emergency Operations Center (EOC) in January 2020, enabling effective coordination with state public health laboratories and other partners to respond to COVID-19 [13]. Using a dedicated EOC email, state NBS programs were able to share information with APHL about reagent and consumable shortages that threatened their ability to continue screening newborns during the pandemic.

## 3. Results

### 3.1. National Technical Assistance Center

The global COVID-19 pandemic made immediate and lasting impacts on NBS laboratory and follow-up operations, requiring programs to make necessary pivots and to identify and implement solutions with enduring impacts. The NewSTEPs website tracked these changes and maintained a resource center for state NBS programs to benefit from the experience of their peers.

Data deposited at the NewSTEPs website revealed that the NBS system was impacted at nearly every touchpoint. Routine communications by NewSTEPs staff with state newborn screening programs verified this observation [14].

Pre-Analytic Impacts: Birth, Specimen Collection, and Courier:Early discharge from hospitals impacted the time of specimen collection. The national recommendation for newborn screening dried blood spot specimen collection is 24–48 h after birth [15]. It was noted that many birth hospitals were discharging healthy mothers and babies before this timeframe to reduce potential for exposure and due to limited space in maternity wards at hospitals;Specimen quality was impacted due to burdens on nursing staff and redistribution of responsibilities as well as due to staff shortages;Courier issues severely limited deliveries of specimens from birthing centers to public health laboratories during certain points in the pandemic. During the 2020 election, an increase in mail-in ballots placed especially burdensome delays on the United States Postal Service. All states rely on specimen transport services to ensure continuity of newborn screening [16];Decreased courier pick-ups, as well as changes in courier pick-up locations, at hospitals further compounded specimen collection and transport activities.

Analytic Impacts: Staff Continuity of Operations:Biosafety of specimens became a concern, with protocols required around handling dried blood spot cards that may have potentially been exposed to infected blood;Laboratory workflows were impacted by the necessity to (1) perform time-critical testing with limited staff; (2) stagger shifts to reduce the potential for exposure within the laboratory workforce; (3) observe physical distancing in the laboratory to maintain staff safety;Managing the impact on staffing due to daycare and school closures was a compounding issue as the pandemic permeated other aspects of day-to-day life in the US;Follow-up staff members were able to transition to working remotely, with laboratory staff required on-site;Program managers integrated planning for the possibility of multiple staff out of work or under quarantine.

Analytic Impacts: Second or Repeat Screens:Families’ refusal to return to the hospital/birthing facility for repeat/second screen specimen collection and confirmatory testing required modifications to protocols to err on the side of testing all specimens regardless of quality;Hospitals/birthing facilities practicing physical distancing and turning away “non-essential” patients impacted the timeliness of subsequent specimen receipt and confirmatory testing;Some outpatient laboratories and clinics closed, impacting the necessity to evaluate all results without the confidence that repeat and confirmatory tests would be conducted in a timely fashion.NBS programs incorporated age-related cut-offs to account for early specimen collection;NBS laboratories initiated or, in some cases continued, screening unsatisfactory specimens and reporting results accordingly;Disorder specific nuances became more pronounced, especially when screening for time-critical versus time-sensitive disorders;An increased reliance on electronic results reporting enabled more efficient information transfer.

Education and Outreach Impacts:Increases in home births required ongoing midwife education, typically using electronic training resources;Telehealth for continuity of follow-up and clinical services became a viable alternative to in-person services;Genetic consultations and access to specialists became more challenging due to COVID-19-related competing priorities across healthcare systems;Increases in education about newborn screening to the extended healthcare community became necessary.

Supply Shortages:Personal Protective Equipment (PPE) for laboratory and hospital staff was limited in quantity at the start of the pandemic (2020);Reagents, testing kids, and other laboratory supplies became impacted by pervasive supply chain issues throughout the pandemic.

### 3.2. Topical COVID-19 and NBS Continuity Webinar Solutions

APHL coordinated with state NBS experts around the US to quickly develop and host national webinars on topics that became relevant at the time as the pandemic and its impacts were being felt across the US NBS programs [9].

An early webinar in May 2020 facilitated by HRSA as a listening session between the National Center for Hearing Assessment and Management (NCHAM) and APHL featured timely information on challenges, barriers, and solutions regarding blood spot screenings, hearing screenings, and parent engagement. The success of this listening session resulted in additional webinars (Table 2) that were coordinated by NewSTEPs and led by speakers from states around the country and able to offer solutions to their peers on pandemic-related issues that were straining the NBS system. Each webinar provided time for robust discussion amongst attendees, creating an open forum for collaboration and ideas exchange to discuss solutions to novel problems.

Webinar topics mirrored the emergence of issues across the nation, starting with the hesitancy of parents to return to the hospital for repeat screens fearing exposure, to concerns by laboratory staff on safe handling of blood spot specimens that may have been exposed to the virus, to navigating solutions for remote work and onboarding staff in a remote environment, to addressing shortages in consumables required to perform testing. These webinars are ongoing [9].

### 3.3. Public Health Emergency Funding

In January 2021, seven NBS programs ratified contracts with APHL to begin funded work on projects with a public health emergency focus, and specifically aimed toward improving NBS operations during the COVID-19 pandemic.

California utilized the funding to develop a comprehensive, virtual, interactive virtual site visit module that allowed for continuity of compliance assessments and education and support for California’s 246 birth facilities. The funding was timely because the COVID-19 pandemic caused the suspension of mandated in-person triennial site visits, which have proven to be essential for assessment to ensure the integrity of birth facilities’ NBS specimen chain of custody processes;Colorado utilized the funding to strengthen and add structure to the NBS follow-up program by expanding the availability of educational materials for stakeholders and formalizing outreach. Within Colorado, the COVID-19 pandemic highlighted how crucial it is for NBS programs to possess the technological capability to collaborate and to utilize tools that are essential to take necessary actions to achieve goals of early detection, despite pandemic challenges;Washington, DC, utilized the funding to design an NBS care model that implements immediate telemedicine for NBS, providing virtual telemedicine options incorporating community-based and patient feedback for optimal clinical and educational interventions;Georgia utilized the funding to implement filter paper testing to avoid social contact required for laboratory blood draw during the global pandemic after the COVID-19 related closure of a genetics lab for biochemical testing in the state. The program was able to maintain uninterrupted continuity of monitoring of blood enzyme concentrations for an NBS disorder that requires comprehensive and continued long term follow-up and treatment;Puerto Rico utilized the funding to procure scarce PPE, which was essential for the continuity of safe testing protocols during the COVID-19 global health crisis;Tennessee utilized the funding to address the COVID-19 related issue of parents being reticent to bring their newborn infants to hospitals for repeat screens necessary to clarify a potentially false-positive result from a newborn screen. To reduce the number of false positives and simultaneously alleviate parental anxiety created by the pandemic, the Tennessee NBS program implemented a second-tier, in-house test for a subset of disorders;Virginia utilized the funding to hire additional NBS scientists for on-call coverage, as well as NBS data entry/verification staff to alleviate the staff shortages caused by symptom monitoring policies for COVID-19, sicknesses, and reallocation of human capital to support COVID-19 testing within the department of health.

Five of the seven funded projects were completed in July 2021, with two projects ongoing with expected completion dates of June 2022.

### 3.4. Telehealth Expansion

The COVID-19 pandemic placed tremendous pressure on the NBS system, with telehealth identified as a useful mechanism to support the continuity of operations. Telehealth solutions should be explored further due to their potential to strengthen the newborn screening system and increase access to newborn screening services, particularly in medically underserved populations. The value of telehealth in the NBS laboratory and follow-up programs is described in a resource developed collaboratively between APHL and AMCHP in February 2021 titled Newborn Screening Continuity of Operations in a Pandemic [17]. Telehealth utility from the provider and clinical perspective is described elsewhere in the literature [18].

Telehealth solutions in NBS laboratory and follow-up programs include the ability to perform remote data analysis, participate in online training, facilitate regular communication between NBS teams and stakeholders despite physical distance, and allow for the reduction in paper-based records in favor of electronic records. The incorporation of electronic results reporting and telehealth solutions can lend to a more durable NBS system.

### 3.5. Survey Results

Responses to the survey fielded by APHL in November 2020 were outlined in a report distributed to state NBS programs titled Impacts to State Newborn Screening Programs from SARS-CoV-2 Pandemic [19]. State NBS programs described a broad spectrum of effects across NBS programs on staff, information technology resources and support, specimen transport, reagent and/or supply shortages, and support from vendors. A key question in the survey asked how programs worked to ensure that NBS remain a priority during the COVID-19 pandemic. Respondents were instructed to select all that apply. In total, 45 respondents provided the following answers: through communication with leadership staff (36); through continuity of operations planning (27); by establishing NBS as an essential service (26); by continuing to engage external stakeholders (24); through policy and advocacy (13); NBS is already considered an essential service (5); communication with human resources and/or clinics and primary care providers (2).

### 3.6. Consumable Shortages Addressed

As the pandemic continued a national laboratory shortage of pipette tips became apparent [20]. In April 2021, APHL leadership worked with the US Department of Health and Human Services (HHS) Testing and Diagnostics Working Group (TDWG) to designate pipette tips for newborn screening as a market priority [21] with the intent of ensuring the continuation of the NBS program in the US throughout the public health emergency [22].

### 3.7. Data Analysis

APHL performs data collection using harmonized newborn screening metrics that measure quality practices within state programs for pre-analytic, analytic, and post-analytic activities [12]. These data are entered into the NewSTEPs Data Repository by states and are used to support data-driven outcome assessments.

For the majority of Quality Indicators being reviewed, in 2020, fewer state NBS programs provided data to the NewSTEPs data repository than in years prior, ostensibly due to the competing priorities resulting from the COVID-19 pandemic. We hypothesized that the greatest negative impacts on quality practices would be realized in the metrics that capture the number of newborns lost to follow-up from borderline screening results, due to closed provider offices and an increased reticence toward returning to clinics for repeat screenings. We also hypothesized that timeliness metrics would suffer, with the process from specimen collection through reporting taking longer in 2020 due to the additional strains resulting from courier delays and NBS program staff shortages during the pandemic.

NewSTEPs Quality Indicator data (Table 3) are summarized below. Caveats of these data summaries are that the states submitting data between 2017 and 2020 varied. As such, only descriptive data are presented. A manuscript is under preparation to fully evaluate the significance of these data.

Timeliness medians did not suffer in 2020, compared with 2017, 2018, and 2019. However, there were data from individual states that showed some significant delays in timeliness, compared with prior years, but this was in a small number of states reporting data and did not impact the overall median;The median percent of specimens that were unacceptable did not increase in 2020 compared with 2017, 2018, and 2019 in states reporting data;The median percent of specimens with missing essential information increased in 2020 compared with 2017, 2018, and 2019 in states reporting data;The median percent of infants that were lost to follow-up after the receipt of an unacceptable dried blood spot specimen increased in 2020 as compared with 2017, 2018, and 2019 in states reporting data;The median percent of infants that were lost to follow-up after an out-of-range result increased in 2020 as compared with 2017, 2018, and 2019 in states reporting data.

### 3.8. State Experiences

A subset of state experiences from APHL member volunteer leaders are described here.

#### 3.8.1. Iowa

Accuracy and quality are vital in every aspect of the NBS process. This point was driven home during the COVID-19 pandemic. Newborn screening programs across the country saw an increase in poor-quality samples, as well as an increase in inaccurate and/or incomplete essential information on the dried blood spot collection form. The increase in poor-quality samples and missing/inaccurate information was a by-product of the redeployment of laboratorians and nurses experienced in the NBS process to other pandemic-related duties. Disruptions of the NBS process were expected during the early months of the pandemic. However, programs are reporting a persistent trend of higher than normal poor-quality samples and missing information and/or errors on the dried blood spot card even now. The persistence of these problems deserves evaluation. Electronic messaging of the essential information from the blood spot card could help reduce problems with missing information and errors. However, this does not address the entire problem.

Overall, a renewed focus on education is key to reducing this ongoing issue. First, NBS programs need to re-engage and reinforce relationships with birthing facilities and remind them how critical staff education is to the overall success of the screening process. Although on-site education will remain an important component of an NBS program’s educational efforts, NBS programs must find a way to make education to birthing facilities and midwives available on-demand, easily accessible, and in an electronic format that is user friendly. Screens are collected 24 h a day, and education should accordingly be made available to accommodate that demand.

#### 3.8.2. New York

The pandemic impacted New York early on and many programmatic changes were made to ensure continuity of operations. NBS follow-up program staff, along with some data entry staff, transitioned to fully remote work. Laboratory staff were trained to analyze results remotely to prepare for a situation where a very limited staff would be available to perform testing. A protocol was created to handle specimens that were collected from babies of known or possible COVID-19 infected mothers. A data review was conducted to change to a risk-based report such that providers who may not have been as involved in newborn screening follow-up could use professional judgment to collect repeat specimens and/or complete diagnostic evaluations. Blast emails were sent to providers with educational materials for specimen collection and follow-up expectations so tools would be in hand for providers who may previously have had limited interaction with the NBS system. All Specialty Care Center Directors were asked to update contact information for themselves and for their staff.

Related to dried blood spot specimen quality, the New York State program has been testing and studying results from suboptimal specimens for several years. Prior to implementing this practice, any specimen that was not optimal was not tested, and a repeat specimen was requested. Following an internal multi-year study of testing initial suboptimal and their repeat specimens, the New York NBS program decided to test and report results for these specimens, making only quantity insufficient and blank cards truly unsatisfactory, requiring a repeat specimen. Up until the COVID-19 pandemic, referrals were only made for analytes at emergency levels and a repeat specimen was requested. During the pandemic, however, the decision was made to refer infants according to normal cut-off values, while still requesting a repeat specimen. This change would reduce the risk of instances in which the repeat specimen was not collected due to the implications of the COVID-19 pandemic. In these situations, the infant’s results were available to the medical community with instruction that the baby is evaluated, diagnosed, and, if necessary, managed.

#### 3.8.3. Texas

Historically, funding sources for Texas newborn screening have been limited. Improvement activities including implementation of new conditions and instrument upgrades were frequently delayed or discontinued due to shortages in funding or other resources. With the pandemic and increasing awareness and recognition of the importance and necessity of public health, federal as well as state funding opportunities to support COVID-19 testing and epidemiology functions and other public health initiatives became more readily available. Infrastructure improvements in health informatics, use of cloud computing services, and information technology solutions for virtual meetings, collaboration, and file sharing made possible by the additional funding modernized and benefited the NBS laboratory and follow-up program. To better prepare for future infectious disease surveillance and outbreak response, efforts were made to increase testing capacity and capability by updating analytical technologies, purchasing next-generation sequencing equipment, hiring and training new technologists, and developing in-house bioinformatics systems, which have been utilized for expansion of NBS genomic sequencing in Texas. To minimize delays in testing results, the Texas NBS staff have been working 6 days a week and most of the holidays since 2006 without additional compensation. As the Texas Department of State Health Services (DSHS) Laboratory started providing COVID-19 testing 7 days a week to meet the urgent demands during the early phase of the pandemic, a new policy to improve employee retention was implemented to provide shift differential pays to compensate laboratory staff working on weekends and holidays, including all NBS staff. Despite all the disruptions and impacts brought upon by the COVID-19 pandemic, the Texas DSHS Laboratory was able to successfully leverage these rare funding opportunities to make changes that can fundamentally benefit all public health testing services and, more specifically, newborn screening.

## 4. Conclusions

A recurring theme emerged via the various mechanisms utilized to understand the impact of the COVID-19 pandemic described in Understanding the Impact of the COVID-19 Pandemic on NBS (survey, telehealth resource development, national webinars, state experiences, public health emergency funding). Data collected for this report pointed to the toll the pandemic took on the public health workforce. Every touchpoint in the NBS laboratory and follow-up system reverberated with the importance of a physically and mentally healthy, agile, and present workforce. The persistent implications of workforce shortages and the impact on the workforce while navigating the COVID-19 pandemic and continuing to perform the essential functions required of NBS programs should be explored further. Protection of the physical and mental well-being of NBS program staff should be pursued as a system-wide imperative and should be incorporated with intention into contingency plans.

Up until the COVID-19 pandemic, state NBS contingency plans had not incorporated planning for scenarios where the entire NBS system across the nation would be impacted in tandem. This required solutions to be created, and communicated, quickly in response to emerging issues that had not previously been considered. The existing strong national network coordination through APHL, in concert with federal, state, and national partners, was a compelling force in information collection and dissemination. Globally, similar systems incorporating multi-sector and multi-stakeholder coordination could benefit regions performing newborn screening outside of the US.

Despite the challenges, the NBS laboratory and follow-up system persevered in the timely screening of nearly every baby born in the US during the COVID-19 pandemic. Data on newborn screening case diagnoses and outcomes are collected with a two-year lag to accommodate for a comprehensive analysis of all available data regarding presumptive positive and confirmatory results. An in-depth analysis of national trends from 2020 and 2021 will be published once the final datasets become available.

Solutions implemented to address COVID-19 threats to the NBS laboratory and follow-up system can be adapted to sustain positive outcomes on program activities even after the pandemic impacts have lessened. For example, establishing diverse platforms for providing education to hospital staff and analyzing results from suboptimal specimens can add value to NBS quality practices even outside of a public health emergency. Additionally, pandemic interventions such as the relaxing of telemedicine policies to enhance uptake of telehealth practices have demonstrable successes in the continuity of operations, which would add value even in post-pandemic surveillance and clinical settings. Increased resilience with remote work and staggered work shifts will continue to introduce efficiencies within workflows. Fundamentally, leveraging existing national resource centers such as NewSTEPs to provide support during an unprecedented time of need has tangible value. APHL serves as a neutral arbiter and broker of information to key partners including the media and government, a role that was elevated and apparent during the COVID-19 pandemic [23]. The severe, long-lasting impacts of the pandemic on NBS have resulted in necessary changes that will leave an indelible, continuing influence on the system.

## Figures and Tables

**Table 1 IJNS-08-00028-t001:** COVID-19 history and newborn screening impact timeline.

December 2019	- **Novel coronavirus detected in Wuhan.**
January 2020	- First COVID-19 case detected in the US; - *APHL establishes its Incident Command System and activates its Emergency Operations Center in response to COVID-19;* - *APHL initiates issuing Lab Alerts and coordinating national on a weekly basis to public health laboratory members as a mechanism for real-time and just-in-time updates.*
February 2020	- World Health Organization (WHO) announces official name for COVID-19; - **Community transmission in the US;** - **US Food and Drug Administration (FDA) issues an Emergency Use Authorization (EUA) for a COVID-19 assay;** - *APHL partners with the Association of State and Territorial Health Officials (ASTHO), the National Association of County and City Health Officials (NACCHO), and the Council of State and Territorial Epidemiologists (CSTE) to send a letter to Congress requesting emergency supplemental funding to support the COVID-19 response;* - *APHL secures funding from CDC Foundation to support select public health laboratories by procuring automated extraction platforms.*
March 2020	- Until 13 March 2020, Public Health Laboratories (APHL member laboratories) were the only laboratories authorized to conduct testing outside of the US CDC; - **All states report COVID-19 cases;** - **WHO declares global pandemic;** - **US President declares nationwide emergency;** - **Coronavirus Aid, Relief, and Economic Security (CARES) Act legislation are signed.**
April 2020	- Schools, daycare closure across much of the United States; - **Significant racial disparities of the pandemic in the US begin to be revealed through data;** - **Operation Warp Speed launches;** - **With funding from the CARES Act of 2020, CDC Epidemiology and Laboratory Capacity for Prevention and Control of Emerging Infectious Diseases (ELC) awards USD 631 million to 64 recipients to support COVID-19 response, including all 50 US states and territories;** - * Newborn Screening (NBS) Laboratories nationwide initiate physical distancing policies and staggered shifts as viral transmission mitigation strategies; * - *APHL began conducting a weekly survey of up to 100 state, local, and territorial public health laboratories to understand the testing capability and capacity for SARS-CoV-2.*
May 2020	- * Initiation of APHL National NBS Webinars as a forum for information exchange; * - **CARES Act Coronavirus Relief Funding legislation is signed;** - **US Coronavirus death toll surpasses 100,000.**
June 2020	- **US Coronavirus cases reach 2 million.**
July 2020	- *APHL announces collaboration with Apple, Google, and Microsoft to enable public health agencies to provide privacy-preserving and user-controlled exposure notifications.*
September 2020	- *APHL releases funding opportunities for states to Address the Impact of the Public Health Emergency on NBS*.
October 2020	- APHL launches the APHL-CDC COVID-19 Associate Program to fill critical roles at all levels of public health laboratory response; - *APHL begins procurement of USD 5 million in equipment to support COVID-19 testing in public health laboratories.*
November 2020	- * APHL launches NBS COVID-19 Survey and issues subsequent report on findings. *
December 2020	- US Food and Drug Administration issues an Emergency Use Authorization for the first COVID-19 vaccine.
January 2021	- The one-year anniversary of the CDC COVID-19 pandemic response in the US; - **US Coronavirus death toll surpasses 400,000;** - * Seven NBS programs initiate projects to improve NBS operations during the COVID-19 pandemic * *;* - *APHL issues a white paper on Smart Testing for Optimizing Pandemic Response;* - *APHL partners with MASCON to procure pipette tips for public health laboratories during a nationwide pipette tip shortage.*
February 2021	- US Coronavirus death toll surpasses 500,000; - *APHL collaborates with the Association of Maternal and Child Health Programs (AMCHP) to publish guidance on NBS Continuity of Operations in a Pandemic, focused on Telehealth*.
March 2021	- US surpasses 100 million vaccines administered for COVID-19; - *APHL launches COVID-19 Genomic Data Specialist program to assist with public health laboratory genomic data and bioinformatics needs;* - *APHL announces the ability to connect Abbott’s BinaxNOW COVID-19 rapid antigen test kits results to public health agencies across the country.*
April 2021	- US surpasses 200 million vaccinations administered for COVID-19; - *US Health and Human Services (HHS) Testing and Diagnostics Working Group (TDWG) issues a memo designating pipette tips for NBS as a national market priority*.
May 2021–December 2021	- APHL continues to collect data from up to 100 public health laboratories on SARS-COV-2 capability, capacity, and ongoing needs. Data collected indicate that since February 2020, public health laboratories have tested over 21 million specimens; - * Monitoring supply chain issues, including reagent and consumables shortages, for continuity of newborn screening testing operations nationally; * - * Monitoring emerging variants of concern and the impact they may have on the public health system, and ongoing efforts to maintain continuity of newborn screening as an essential service * *;* - *APHL continues to provide member service in support of public health laboratory response to the ongoing COVID-19 pandemic.*

Bold font: event external to APHL; *Italics font*: APHL-specific event; *Red font*: NBS-system-specific event.

**Table 2 IJNS-08-00028-t002:** COVID-19 and newborn screening topical webinars hosted by APHL.

Topic	Date	Presented By	Key Takeaway
Screening of Unsatisfactory Specimens	8 May 2020	Tennessee, Maryland	During the COVID-19 pandemic a number of staff shortages within the clinical healthcare system, coupled with the risk of exposure to the virus, greatly lessened the number of visits that were being made to hospitals for non-essential services. NBS programs experienced increased hesitancy of families with newborns to return to the clinic for the collection of repeat blood spots for newborn screening. Response to the limitations of re-testing included screening and reporting results on unsatisfactory initial specimens to ensure that each newborn was screened a minimum of one time.
Newborn Screening COVID-19 Challenges and Response	21 May 2020	HRSA, APHL, NCHAM, New York, North Carolina, Genetic Alliance, Hands and Voices	Challenges, barriers, and solutions to dried blood spots and newborn hearing screening, as well as family engagement perspectives during the COVID-19 pandemic in the US.
Telehealth in Newborn Screening	22 May 2020	Hawaii, Minnesota	Telehealth initiatives were utilized for continuity of operations by newborn screening in response to the COVID-19 pandemic.
Biosafety	28 May 2020	Centers for Disease Control and Prevention (CDC)	Addressing biosafety of dried blood spot specimens in response to the COVID-19 pandemic requires data on the viability of the virus on specimens from mothers or babies who are COVID-19 positive.
Staffing and Telework	11 June 2020	Iowa, APHL	Addressing staffing and telework during the COVID-19 pandemic requires the following considerations: access to equipment; permission for telework; connectivity; effective communication; productivity; loss of in-person team environment; ethical concerns; staff disparities in ability to telework; minimal needs for robust telework, such as a business phone line, video capabilities, e-faxing, secure email.
Virtual Engagement	9 July 2020	Genetic Alliance, Virginia, Tennessee, North Dakota, New York	Addressing virtual engagement in response to the COVID-19 pandemic is an ongoing imperative. Conference applications and planning software can be instrumental in planning virtual events.
Electronic Reporting	12 August 2020	Nevada, Louisiana, Texas	The COVID-19 pandemic required programs to reduce NBS results reporting using paper and transition to electronic results reporting and messaging. Privacy concerns and data security challenges must be identified and tackled to implement electronic reporting.
Resource Shortages and Staffing Limitations	5 October 2020	Open Discussion	Discussion among all state NBS programs participants around individual experiences with resource shortages and staffing limitations during the COVID-19 pandemic.
Resources and Best Practices for Remote Follow-Up Work	17 December 2020	Open Discussion	Discussion among all state newborn screening program participants around the latest practices and protocols for performing follow-up from home during the COVID-19 pandemic.
Resources and Best Practices for Managing Staff during COVID-19	28 January 2021	Open Discussion	Geared toward supervisors of NBS programs. NBS laboratory and follow-up staff shared strategies for training employees, supporting remote staff, and how to manage employee productivity and accountability during the COVID-19 pandemic.
Staff Training and Onboarding During COVID-19	16 February 2021	Open Discussion	Geared toward supervisors and staff in charge of training newborn screening laboratory and follow-up staff, participants shared experiences and lessons learned around training new staff both virtually and on-site during the COVID-19 pandemic.
Cleaning Methods for Pipette Tips	23 March 2021	New York, Maryland	Cleaning methods for pipette tips as a short-term solution to address supply limitations, addressing method validation and concerns around cross-contamination.
Building More Resilient Newborn Screening Systems	7 April 2021	APHL	Information sharing regarding continuous quality improvement (CQI) tools and strategies for developing more adaptive and resilient newborn screening systems.
Contingency Planning	16 December 2021	South Carolina, Louisiana, Iowa	Discussion of newborn screening contingency planning and lessons learned from state experiences in addressing emergency situations and the use of Continuity of Operations Planning (COOP) during the COVID-19 pandemic.

**Table 3 IJNS-08-00028-t003:** NewSTEPs quality indicator data national summary, 2017–2020.

NewSTEPs Quality Indicator	Year	Number of States Reporting (Births Represented)	Median	NewSTEPs Quality Indicator	Year	Number of States Reporting (Births Represented)	Median
Percent of Dried Blood Spot Specimens that were Unacceptable (Unsatisfactory)	2017	34 (3,150,513)	1.47	Percent of Infants with No Resolution Following an Out of Range Result from a Dried Blood Spot Specimen	2017	6 (760,307)	1.32
	2018	33 (2,725,317)	1.63		2018	9 (1,151,034)	1.5
	2019	29 (2,522,850)	1.6		2019	11 (1,306,753)	1.22
	2020	23 (1,731,157)	1.12		2020	13 (1,139,682)	2.42
Percent of Dried Blood Spot Specimens Missing Essential Information	2017	30 (2,932,360)	1.8	Percent of Dried Blood Spot Specimens Collected within 48 Hours of Birth	2017	34 (2,998,524)	94.18
	2018	32 (2,690,628)	1.22		2018	34 (2,796,433)	94.62
	2019	28 (2,489,112)	1.39		2019	28 (2,396,738)	96.26
	2020	14 (1,233,014)	2.1		2020	20 (1,522,389)	97.75
Percent of Eligible Newborns Not Receiving a Dried Blood Spot Newborn Screen	2017	9 (1,088,165)	0.49	Percent of Dried Blood Spot Specimens Received at NBS Laboratory within 24 h of Collection	2017	34 (2,998,524)	36.47
	2018	11 (1,012,836)	0.32		2018	34 (2,796,433)	41.36
	2019	13 (1,206,276)	0.29		2019	28 (2,396,738)	48.25
	2020	13 (1,097,817)	0.35		2020	20 (1,522,389)	51.75
Percent of Infants with No Recorded Final Resolution Following Receipt of an Unacceptable Dried Blood Spot Specimen	2017	8 (1,034,884)	3.11	Percent of Time Critical Specimen Results Reported within 5 Days of Birth	2017	23 (2,161,746)	37.93
	2018	10 (1,213,919)	2.12		2018	24 (2,287,170)	47.82
	2019	13 (1,405,017)	2.77		2019	25 (2,249,817)	45.89
	2020	12 (1,083,658)	4.25		2020	20 (1,593,377)	48.25
Percent of Infants with No Recorded Final Resolution Following a Borderline Result from a Dried Blood Spot Specimen	2017	5 (288,649)	0.83	Percent of Non-Time Critical Specimen Results Reported within 7 Days of Birth	2017	24 (2,172,191)	74.08
	2018	8 (924,796)	2.43		2018	24 (2,287,170)	69.33
	2019	11 (1,120,609)	1.03		2019	24 (2,223,498)	61.02
	2020	12 (1,106,234)	1.58		2020	20 (1,549,528)	69.59

## Data Availability

The data presented in this study are available on request from the corresponding author. The data are not publicly available due to data security and privacy outlined in memorandums of understanding between APHL and state public health laboratories.

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
