# Peer review of "COVID-19 Pandemic-Related Impacts on Newborn Screening Public Health Surveillance"

_2409-515X, 2022, doi:10.3390/ijns8020028_

Round 1
Reviewer 1 Report
No comments
Author Response
Thank you for your thoughtful review.
Reviewer 2 Report
This manuscript deals with the important topic of newborn screening (NBS), summarizing several activities implemented in the USA, aiming to maintain the quality of NBS. Even if the authors describe their own experience on the US-based steps taken to achieve this goal, the problems shown (Lines 141-201) and the possible solutions developed are universal and international. This, together with the convincing way they are presented are major strengths of the manuscript.
Similar challenges due to COVID had to be faced in many (if not all) other NBS laboratories (including ours at a completely different location). Moreover, many of the solutions described in this manuscript have been used in our laboratory too. Nevertheless, it would have had probably been beneficious for other labs (including ours) to have such a comprehensive system like the NewSTEPs of the APHL.
NBS labs may have different problems depending on numerous factors such as their location, disorders screened etc. The challenges have also been continuously changing throughout this period, which required all NBS labs to re-think their daily routine and emergency scenarios. It is important to note that the system described in the manuscript allowed the labs to prioritize according to their own needs, as exemplified by Sections 3.3 and 3.8.
A minor comment: The authors should include a paragraph on the need and the possibilities to develop similar systems that are more specific to non-US regions like Europe, Asia etc. Otherwise, I would recommend accepting the manuscript in its present form.
Author Response
Thank you for your thoughtful review. We have added a few lines on the benefits of developing similar systems as described in the US for a global NBS community. These can be found on lines 423-425 of the final manuscript.
Reviewer 3 Report
A clearly written description of consquences of the Covanid 19 epidemic for newborn screening in the US and measures taken by the local screening laboratories and the Association of Public Health Laboratories to ensure the continuation and quality of the newborn screening programs.
Author Response
Thank you for your thoughtful review.